# General Purpose Pharmacokinetic-Pharmacodynamic Models for Target-Controlled Infusion of Anaesthetic Drugs: A Narrative Review

**DOI:** 10.3390/jcm11092487

**Published:** 2022-04-28

**Authors:** Ophélie Vandemoortele, Laura N. Hannivoort, Florian Vanhoorebeeck, Michel M. R. F. Struys, Hugo E. M. Vereecke

**Affiliations:** 1Department of Anaesthesia and Reanimation, UZ Leuven, 3000 Leuven, Belgium; ophelie.vandemoortele@student.kuleuven.be; 2Department of Anesthesiology, University of Groningen, University Medical Center Groningen, 9713 GZ Groningen, The Netherlands; l.hannivoort@umcg.nl (L.N.H.); m.m.r.f.struys@umcg.nl (M.M.R.F.S.); 3Department of Emergency Medicine, ZNA Middelheim, 2020 Antwerpen, Belgium; florian.vanhoorebeeck@zna.be; 4Department of Basic and Applied Medicine, Ghent University, 9000 Gent, Belgium; 5Department of Anaesthesia and Reanimation, AZ Sint-Jan Brugge-Oostende AV, 8000 Brugge, Belgium

**Keywords:** propofol, remifentanil, dexmedetomidine, general purpose pharmacokinetic-pharmacodynamic models, target controlled infusion

## Abstract

Target controlled infusion (TCI) is a clinically-available and widely-used computer-controlled method of drug administration, adjusting the drug titration towards user selected plasma- or effect-site concentrations, calculated according to pharmacokinetic-pharmacodynamic (PKPD) models. Although this technology is clinically available for several anaesthetic drugs, the contemporary commercialised PKPD models suffer from multiple limitations. First, PKPD models for anaesthetic drugs are developed using deliberately selected patient populations, often excluding the more challenging populations, such as children, obese or elderly patients, of whom the body composition or elimination mechanisms may be structurally different compared to the lean adult patient population. Separate PKPD models have been developed for some of these subcategories, but the availability of multiple PKPD models for a single drug increases the risk for invalid model selection by the user. Second, some models are restricted to the prediction of plasma-concentration without enabling effect-site controlled TCI or they identify the effect-site equilibration rate constant using methods other than PKPD modelling. Advances in computing and the emergence of globally collected databases has allowed the development of new “general purpose” PKPD models. These take on the challenging task of identifying the relationships between patient covariates (age, weight, sex, etc) and the volumes and clearances of multi-compartmental pharmacokinetic models applicable across broad populations from neonates to the elderly, from the underweight to the obese. These models address the issues of allometric scaling of body weight and size, body composition, sex differences, changes with advanced age, and for young children, changes with maturation and growth. General purpose models for propofol, remifentanil and dexmedetomidine have appeared and these greatly reduce the risk of invalid model selection. In this narrative review, we discuss the development, characteristics and validation of several described general purpose PKPD models for anaesthetic drugs.

## 1. Introduction

Target controlled infusion (TCI) is a clinically available and widely used computer controlled method of drug administration for intravenous drugs, in which the initial bolus dose and the subsequent infusion rates of a drug are adapted according to population-derived pharmacokinetic-pharmacodynamic (PKPD) models. First, the pharmacokinetic (PK) model quantifies the expected time course of the plasma concentration of a drug, as determined by analysing repetitive measurements of plasma concentrations in a specific study population of patients or healthy volunteers. Individually observed measurements of drug concentration are pooled together whereafter non-linear analytical software is used to extract the best fitting mathematical equation that describes the population average time course of the plasma concentration of the drug. For most anaesthetic drugs, the initial structural pharmacokinetic model is defined as a multi-compartmental model with a specific volume of distribution and clearance for each compartment. Incorporating relevant demographic and/or physiologic covariates into the equations for volumes of distribution or rate constants or clearance does account for predictable sources of biological variability. This implies that a model with accurate covariate selection often will reduce the in-between subject variability in the model predictions of the plasma concentration compared to a model without, or with suboptimal, selection of demographic covariates [1,2,3].

In addition to the PK model, a pharmacokinetic-pharmacodynamic (PKPD) model is developed by defining an additional drug concentration in a theoretical effect-site compartment that is linked to the plasma concentration with a single equilibration rate constant (k*_e*0*_*). This is a mathematical solution to compensate for the commonly observed time delay between a change in plasma concentration and the onset of the subsequent clinical effect. A change in the effect-site concentration will therefore better correlate with the observed time course of the clinical effect, compared to changes in the plasma concentration over time.

In clinical practice, drug administration through TCI requires a computer-controlled infusion pump, for which input of the demographic variables is required first. Thereafter, the anaesthesiologist sets a desired plasma or effect-site concentration as primary target. Rather than calculating the bolus dose or infusion rate by heart, in a linear weight-adjusted fashion, the volumes of distribution and clearance rates that are defined by the PKPD model (using both linear and non-linear mathematics) will determine the initial bolus dose and subsequent adaptations in the drug administration rate to reach and maintain a steady state in either the plasma- or the effect-site concentration. The input of individual demographic variables in the models’ equations further adds an individual factor to the drug administration after other random sources of biological variability are accounted for. Despite this incorporation of individual demographic data, the final predicted target concentration remains a representation of a population-average PKPD behaviour of the drug. As it is obvious that not all patients behave according to the average of a historical population, one may still need to adjust the initially set target concentration towards a steady state condition that better matches the desired clinical effect. Despite this latter limitation, a TCI-controlled titration of an anaesthetic drug provides more reproducible clinical conditions during induction, maintenance and recovery of anaesthesia compared to manually administered intravenous drugs.

The contemporary commercialised PKPD models for anaesthetic drugs, such as propofol and remifentanil, were developed within the demographic constraints of the original historical study population, either being healthy volunteers or patients, and often excluding the challenging populations, such as the elderly, children or obese patients of whom the body composition or elimination mechanisms may be structurally different compared to the lean adult patient population (Table 1). Consequently, these models were developed using populations with limited demographic variability and may have failed to identify relevant covariates that result in improved prediction performance of the model when applied in these challenging populations. Due to the high number of smaller studies, there may be multiple models available for a single drug, and the clinician needs to decide for every case whether the selected PKPD model is sufficiently representing the demographic characteristics of the patient.

One approach to solve some of the limitations is to combine historical databases from multiple studies into one large, globally collected database which contains the PKPD information of commonly used anaesthetic drugs in diverse patient populations. The open TCI initiative is an example of such an endeavour. This pooled dataset opened up new opportunities to construct so-called “general purpose” PKPD models [2,3,4,18]. Apart from the expected broader applicability, these PKPD models also include innovative modelling methods (such as allometric scaling [19]). Allometry enables the incorporation of covariates relating to size in such a way that extrapolation of covariates between diverse populations becomes more biologically plausible. This may appear especially valuable to define subjects at the extremes of the demographic spectrum or for patient populations for whom only scarce data are available in the database.

This narrative review aims to describe “general purpose” models of anaesthetic drugs, and reviews the innovations in their respective covariate selection. We also explore the availability of prospective validation results. The focus of the review is restricted to models that are currently applied in a commercially distributed TCI pump, or for which, to the best of our knowledge, a commercial applicability will be available soon. However, as the commercialisation of technology is a continuous and non-exhaustive process, the authors do not claim completeness. However, we found that the models included in this review provide a sufficient illustrative framework to point out innovations and limitations of the “general purpose” PKPD technology.

## 2. Comparing the Performance of Multiple PKPD Models

The accuracy of a PKPD model can be prospectively validated by comparing the model’s predicted plasma concentration with measured drug concentrations in a number of consecutive blood samples, and demonstrating acceptable ranges of error over time [20,21]. As the effect-site concentration is a virtual concept, the accuracy of predicted effect-site concentration requires an indirect validation by demonstrating similar time courses between a measured drug effect and the corresponding predicted effect-site concentrations [22].

Inspired by the earlier work of Sheiner and Beal, Varvel et al. proposed criteria for the predictive performance of TCI models, which can be used to compare the performance of different PKPD models: the median performance error (MDPE), median absolute performance error (MDAPE), divergence, and wobble [23].The prediction error is the deviation between each measured and predicted concentration at one time point. The MDPE represents bias of the model predictions, which is the tendency to systematically over- or underestimate the measured values. Ideally, the MDPE should approach 0 as close as possible. The MDAPE represents the precision of the model predictions and should therefore be as small as possible. For plasma concentrations, an MDAPE below 20% is commonly considered to be clinically acceptable. Divergence reflects the performance stability over time. It is defined by the slope of the linear regression of the performance errors plotted against time. A positive value indicates the progressive widening of the gap between predicted and measured concentrations, whereas a negative value reveals a convergence towards smaller errors over time. Wobble indicates the intrasubject variability of the performance error.

## 3. Limitations of Contemporary TCI Models

Multiple PKPD models have been published to describe the PKPD of the same drug, but they often differ in the final selection of the structural multicompartmental model and added covariates [6,7,8,11,20]. The availability of multiple models for a single drug is a potential source of confusion in clinical practice, as it is not obvious to decide whether a selected model sufficiently represents the patient under the care of the anaesthetic practitioner [20]. The differences between PKPD models are closely related to the demographic characteristics of the studied population and to the applied methodology used for model development. Table 1 provides an overview of the demographic range of the respective study populations of commonly applied PKPD models in commercialised TCI systems. A model that describes the PKPD behaviour of a drug in one population (e.g., lean adults) should not blindly be applied in a population with different demographic characteristics (e.g., obese adults, children etc.), as these patients may have considerably different PKPD characteristics. Additionally, methodological differences between studies may result in the different identification of relevant covariates, such as the timing and method of blood sampling (arterial versus venous), differences in analytical approach and the use of separate studies to develop respectively the pharmacokinetic and the pharmacodynamic model, instead of observing both endpoints in a single time-synchronised setting [1,20]. Some models are therefore restricted to the prediction of plasma-concentration without enabling effect-site controlled TCI.

The effect-measures used to predict effect-site concentrations may also different between PKPD models, which can hamper the comparability of different effect-site concentrations predicted for the same drug. For anaesthetic drugs, a continuous effect measure derived from the frontal electroencephalogram is commonly used as a surrogate for the hypnotic drug effect. However, different electroencephalographic-derived indices were used between different studies. For example, for propofol, the canonical univariate analysis was used during the development of the Schnider model, while the spectral edge frequency was used for the Marsh model (in a separate analysis from the PK model) and the bispectral index for the Modified Marsh model [20]. Apart from electroencephalographic indices, other effects (such as heart rate or mean arterial blood pressure) have also been used to develop separate PKPD models for a single drug [5]. All of these variations increase the risk for misinterpretation by the clinician.

Even when a methodologically sound approach is used for data analysis, suboptimal covariates may be selected to define the final model. The most infamous example of this is the use of lean body mass (LBM), as calculated by the James equation, in the Schnider model for propofol and the Minto model for remifentanil (Table 2) [6,9,10,12,24]. The quadratic function in the James equation results in distorted model predictions, especially in obese patients. For example, the LBM of an obese patient would become unrealistically low, or even in the negative values for the morbidly obese, due to the quadratic functions embedded in the equation [20]. This limitation unfortunately remained unnoticed during model development because the original dataset did not include morbidly obese patients with a BMI above 32 kg⋅m^−2^. New models should therefore avoid covariates that use equations with a risk of unrealistic output when extrapolating the demographic input towards the extremes of the demographic spectrum.

## 4. General Purpose PKPD Models: One Model to Fit Them All?

### 4.1. The Open TCI Initiative

The Open TCI Initiative has been initiated as an online platform (http://www.opentci.org (access date: 20 March 2022)) wherein programmers, engineers and researchers globally connect and share open-source datasets that include an increasing amount of information related to the pharmacokinetics and dynamics of drugs, although it currently primarily focusses on propofol and remifentanil. The ethics and etiquette of data sharing in pharmacokinetic studies has been discussed widely [25]. The open TCI initiative database, with deidentified data, includes demographic characteristics of patients or volunteers, drug doses and infusion rates, measured plasma concentrations and time-synchronised measurements of drug effects. This database, combined with the availability of computers with improved calculation power, serve as a basis for the development of “general purpose” PKPD models for commonly used anaesthetic drugs.

### 4.2. The Eleveld PKPD Model for Propofol

After proposing a first analysis using the data from the open TCI database for propofol, Eleveld et al. published a preliminary PK model in 2014. However, the amount of available data expanded further over time, resulting in a second publication of a “general purpose” PKPD model in 2018 [5,26]. Eleveld et al. combined the open TCI database with new institutional pharmacokinetic datasets, yielding a total of 30 studies (15,433 propofol concentration measurements), of which 5 had simultaneous bispectral index registrations (28,639 BIS datapoints), derived from 1033 individual patients or volunteers. With an age range between 27 weeks post menstrual age (PMA) up to 88 years and a weight range between 680 g up to 160 kg, the Eleveld model is expected to be applicable in a broad population of patients (Table 1).

The structural model is a 3-compartmental mammillary model in which classical demographic variables, such as age, weight, height and sex were identified as covariates to improve the prediction performance of the model (Table 3). Lean body mass (used as a size descriptor in the Schnider model for propofol and the Minto model for remifentanil) has been replaced by fat-free-mass (FFM), calculated using the Al-Sallami formula, which is applicable for both adults and children from the age of three years. Compared to the James equation for LBM, the Al-Sallami equation is less prone for deviating and producing unrealistic results in the morbidly obese patient (Table 2).

As an additional innovation, allometric scaling as described by Kleiber’s law was applied to scale volumes of distribution and clearances of the PKPD model [28]. Kleiber’s law identifies mathematical constants in biology such as the constant relationship observed between total body weight and metabolic rate. The West-Brown-Enquist model further elaborated on Kleiber’s law [29]. In the Eleveld model, volumes of distribution relate to body size with a fixed power of 1, but clearances (based on basal metabolic rate) relate to body size to a fixed power of ¾. Additionally, the elimination constants Cl2 and Cl3 (which results from the two combined equilibration constants (denoted as k) that describe respectively the in and outflow of a compartment) are respectively scaled to the size of volumes of distribution V2 and V3 instead of using a classical body size scale (such as FFM or total body weight). The inclusion of these allometrically scaled functions to define covariates are expected to avoid unrealistic parameter predictions when the model is used on patients with demographic characteristics that range outside the originally studied population, such as the morbidly obese or children and neonates (Table 1).

For (premature) neonates, a maturation factor was identified during the modelling process which is based on post-menstrual age (PMA) as input for calculating intercompartmental and elimination clearance.

The concomitant use of opioids was also identified as a significant covariate to defining the pharmacokinetic model parameters, namely V3 and CL1. As such, when applying the Eleveld model in a TCI pump, the clinician will first be prompted to decide whether to apply the propofol administration with or without opioids. Due to the scarce, heterogenous and imprecise data on opioid use in the original publications, only the presence or absence of opioid use could be incorporated as a covariate, without a clear differentiation between high or low doses of opioids (Table 3).

The pharmacodynamic model of Eleveld used bispectral index (BIS) as the measure of effect, which is widely available for clinical use. As such, propofol can now be titrated towards a predicted Ce_50_ for BIS (= the effect-site concentration that correlates with a BIS of 47 on average in the population), which allows comparison to the individual BIS response. This results in an opportunity to identify the sensitive, normal or resistant responding individual to a propofol administration through measuring the difference between the expected and the measured BIS. The Eleveld model might therefore also become an appealing option for closed loop anaesthesia systems when BIS is used as the hypnotic effect measure.

Another newly identified characteristic of the PD model is the age dependent effect-site equilibration constant k*_e*0*_*, which reflects the progressively lower effect-site concentration of propofol that should be targeted with increasing age in order to reach 50% of maximal effect. For a 20-year-old patient, the Eleveld model predicts a Ce_50_ of 3.4 µg⋅mL^−1^, whereas in an 80-year-old patient, the Ce_50_ is 2.3 µg⋅mL^−1^. Also in the pharmacodynamic model, allometric scaling is applied to scale the equilibration constant *k_e0_* to weight in a ‒0.25 power function. This was done to yield more realistic estimations of the pharmacodynamics in children, even though the fit of the final model was not significantly improved by doing so.

Some of the final Eleveld model characteristics, such as the fixed values of the allometric scaling, remain a source of debate and might still be criticised on theoretical grounds [19]. A prospective validation was therefore performed by Vellinga et al. which provides adequate prediction performance, both in children, the elderly, obese and lean adults [30]. Additionally, it has been shown in simulations that propofol doses and infusion rates administered by means of the Eleveld PKPD model are compatible with the SmPC-advised doses and infusion rates for all age groups [5].

Despite this evidence of high-performance results, showing lower MDPE’s and MDAPE’s compared to other PKPD models, the added value of a “general purpose” model is not to outperform other models, but rather to provide a solution for reducing model selection errors by clinicians using a TCI pump.

### 4.3. The Eleveld PKPD Model for Remifentanil

For remifentanil, a number of PKPD models have been published, but only the model developed by Minto et al. is commercially available [12]. The Minto model is based on a dataset of 65 non-obese adults, aged between 20 to 77 years. The mathematical limitation of the LBM calculation, as observed for the Schnider model for propofol, is also an issue for the Minto remifentanil model. Eleveld et al. therefore developed a new model for remifentanil, merging data from several studies [12,13,31]. Although the collected database was less rich compared to the database for propofol, a more generally applicable PKPD model could be developed, including the ability to use the model for remifentanil titration in children and obese patients (Table 1). The pharmacokinetic covariates in the final model include total body weight and fat free mass (as calculated by Al-Sallami) as metrics for body size. Similar to the propofol general purpose model, the three volumes and elimination clearances are allometrically scaled to fat free mass and the intercompartmental distribution is scaled to the volumes of distribution, (Table 3). Even with a limited number of moderately obese (up to a body mass index of 35 kg⋅m^−2^) and no morbidly obese patients included in the open TCI dataset for remifentanil, the proposed solutions aim for an acceptable model performance for the obese population also (Table 2). Indeed, a prospective bootstrapped validation, comparing the Eleveld remifentanil model with the models of Minto, Egan, La Colla and Rigby-Jones (for infants), revealed an MDPE(between −1.54% to 4.45%) and MDAPE(between 14.8% to 26.9%) in all subgroups. The Eleveld model identified a previously unknown clearance reduction of remifentanil in the youngest children, assumed to be evoked by the immaturity of nonspecific tissue esterase activity. Only 50% of maturation seems to be reached at average birth weight (3.6 kg), and this kept developing up until the age of 14. This finding requires prospective confirmation, preferably by identifying a biochemical mechanism that explains the underlying ethiology. A prospective validation study in populations with different demographic characteristics has not been published yet.

The Eleveld remifentanil pharmacokinetic model is expanded with an effect-site compartment and is therefore suited to drive an effect-site controlled TCI system. However, the pharmacodynamic effect of remifentanil is derived from the spectral edge frequency measurements in the adult population, as pharmacodynamic data from other populations were lacking. Prospective validation of the clinical performance will be required to confirm safe clinical performance of effect-site controlled remifentanil titration, especially in children. In a recent simulation study, target ranges for effect-site controlled TCI with the Eleveld PKPD model are proposed for several subcategories of patients to ensure that remifentanil administration is in concordance with the official dosing advice [32].

### 4.4. The Kim-Obara-Egan PK Model for Remifentanil

Kim et al. combined data from nine studies that partially overlapped with the studies used in the Eleveld model [14]. They did not include children, but rather focussed on the inclusion of a larger proportion of obese and morbidly obese adults. The BMI range of the 229 analysed individuals was 16.1 to 73.7 kg m^−2^. No pharmacodynamic model was added, as there was a lack of EEG-derived measures of drug effect in many patients. A consequence of this pragmatic solution is that the Kim-Obara-Egan model will only be applicable in a plasma-controlled TCI setting.

Despite the independent work of the research groups of Eleveld et al. and Kim et al., both found a very similar three compartmental structural model, with comparable covariates to fit the data best (Table 3). Both found that the increase in total body mass in the obese patients is mainly attributable to a gain in fatty tissues, which is best quantified by FFM. As Kim et al. studied adults only, they chose the equation of FFM described by Janmahasatian (Table 2). The Al-Sallami formula used by Eleveld et al., is based on the Janmahasatian formula, but a modification is added to fit a pediatric population (Table 2). As the database for the Kim-Obara-Egan model included obese and morbidly obese patients, the choice for FFM both by Kim et al. and Eleveld et al. seems to confirm its validity.

The Kim-Obara-Egan model also applies allometric scaling. However, the power to which the volumes and clearances are scaled differ compared to the Eleveld model. The optimal value of the allometric exponent was defined by Kim et al. through extraction of the best fitting model on the dataset after different options for the power parameter were tested. Whether this data extracted approach is superior to the fixed constant value of the allometric exponent remains a matter of debate and probably requires further research [13,19].

### 4.5. The Hannivoort-Colin PKPD Model for Dexmedetomidine

Dexmedetomidine is an alpha-2 adrenoreceptor agonist effective on both presynaptic and postsynaptic receptors, which decreases the sympathetic transmission. It has sedative and analgesic properties, induces bradycardia with increasing doses and has a complex biphasic effect on blood pressure. At low drug concentrations below 2 ng/mL, dexmedetomidine induces hypotension by activating receptors in the central nervous system. Higher concentrations induce a progressively intensifying hypertension through the activation of receptors in the peripheral vascular smooth muscles. However, this biphasic concentration-effect relationship is only valid when the drug is administered in a sufficiently slow infusion speed. Higher administration rates may disrupt this relationship because a faster onset of the peripheral vasoconstrictor effect, may cause significant hypertension and bradycardia, before a slower onset of the centrally mediated hypotensive effect can counter the vasoconstriction.

Dexmedetomidine has become a popular drug in intensive care for continuous sedation and as an adjuvant drug for opioid sparing anaesthetic strategies. The onset time of a continuous infusion of dexmedetomidine is slow, and therefore an anaesthetic practitioner might be tempted to increase the infusion speed above 6 microg/kg/h or administer an initial bolus dose to speed up the onset of effect. TCI administration for dexmedetomidine might therefore be advantageous to reach and maintain a therapeutic effect with improved reproducibility and to ensure both a safe hemodynamic infusion profile when compared to a weight-adjusted bolus and continuous infusion.

Originally published PKPD models for dexmedetomidine, such as the Dyck model as well as the Dutta [33] and Talke [34] model, were often derived from intensive care patients who suffer from co-morbidities or use large numbers of potentially interacting drugs. Moreover, prospective validation of these models shows a tendency to underestimate higher drug concentrations. In 2015, Hannivoort et al. published a new pharmacokinetic model based on a dataset derived from 18 healthy adult volunteers, including the administration of higher concentrations of dexmedetomidine. As this model is based on a small number of adults and elderly volunteers only, it can hardly be considered to be a “general purpose model”. However, several methodological choices were applied during the model development that built further on the lessons learned from the “general purpose” models.

First, the studied population was stratified for sex and age to ensure a similar number of datapoints over a wide range of ages. Both sexes were equally distributed over three age ranges (20 to 35, 36 to 55 and 56 to 70 years). Although morbid obesity was an exclusion criterion in this study, the final dataset did include some information for patients with a BMI of up to 29.3 kg⋅m^−2^. The inclusion of healthy volunteers allowed for the evaluation of the pharmacokinetics without severe co-morbidities or potentially interacting drugs.

Hannivoort found that a three-compartment structural model described the pharmacokinetic data best, but also found that allometric scaling of volumes and clearances resulted in a better fit (Table 3). The Hannivoort PK model performed better than previously published PK models in an initial prospective validation study, where patients received dexmedetomidine during spinal anaesthesia [27]. However, in a subsequent interaction study by Weerink et al., in which healthy volunteers received dexmedetomidine and remifentanil, the Hannivoort PK model performed well only up until targets < 3 ng mL^−1^; but for higher concentrations (which are rarely necessary in clinical practice) non-linear kinetics were suspected and confirmed by Alvarez-Jiminez et al. in a closer analysis of both the Hannivoort and Weerink data [35,36,37].

Although the Hannivoort-Colin dexmedetomidine PK model cannot be considered a general purpose model as it only included 18 adult volunteers, the use of allometric scaling for weight and the stratification of age in the selected population does imply an improved prediction accuracy, even when the demographic characteristics exceed the range of the original studied population. However, it remains a topic for prospective validation whether the model will also perform adequately in children or obese patients. The use of total body weight may hamper performance in obese patients, despite the expected adaptation due to the allometric scaling. The absence of maturation factors in the model likely reduces the usability in (premature) neonates, but dexmedetomidine is at present hardly used in this population.

In 2017, Colin et al. published several pharmacodynamic models derived from the same dataset as the Hannivoort PK model [36,37]. Several clinical effects of dexmedetomidine were modelled in relation to the time course of the plasmaconcentration as predicted by the Hannivoort pharmacokinetic model: BIS (with and without non-noxious or noxious stimulation), the modified observers assessment of alertness and sedation (MOAA/S) score, blood pressure and heart rate.

For BIS, the authors observed a typical rise in BIS after the volunteers were stimulated to determine the MOAA/S score. The effect-site concentration model therefore had to combine both a non-stimulated (before MOAA/S scoring) and a stimulated state. This combined PD model for BIS did predict the time course of a measured BIS, including the rise and gradual fade-out back to the non-stimulated state once a single stimulus or a sequence of stimulations ceases. Due to the dependency of BIS to this unpredictable timing and intensity of stimulation, it may not be a very practical measure for effect-site controlled TCI. Therefore, Colin et al. explored a practical alternative PD model that describes the relationship between the drug concentrations and the probability of reaching one of the scores on the MOAA/S scale. The MOAA/S scale is a categorical score for sedation, that requires a transformation into a continuous logit-scaled cumulative probability scale before PD modelling. The final PD model enables anaesthetic practitioners to target an effect-site concentration that represents the highest probability of reaching a desired MOAA/S score of, for example,3 or less, which represents a moderate sedation level suited for many diagnostic or therapeutic procedures. Before using a commercialised effect-site controlled TCI system for administration of dexmedetomidine, the user needs to inquire first as to which clinical effect is represented by the modelled effect-site concentration.

An accompanying PD model for heart rate and mean arterial pressure (MAP) was simultaneously published to improve the control over the hemodynamic effects of dexmedetomidine. However, as the database only contains observations of effect after respecting an equilibration time with steady plasma concentrations, the final PD model does not predict the undesired hypertensive effect of dexmedetomidine seen after a fast administration.

Heart rate was found to decrease with increasing concentrations. A heart rate < 40 bpm was a stopping criterium in the study for safety reasons, so the predictions of HR below 40 must be considered as hypothetical extrapolations of the observed HR values. For MAP, dexmedetomidine describes the previously mentioned biphasic effect, with decreasing MAP below baseline at lower concentrations (<2 ng/mL) and a subsequent increase in MAP towards hypertensive values at higher concentrations. The Hannivoort-Colin model combines two effect-sites, each representing hypertensive and hypotensive effects, respectively, and characterised by separate equilibration rate constants (k*_e*0*_*) and concentrations of 50% of maximal effect (Ce_50_).

By combining all PD models, the anaesthetic practitioner can now target an effect-site concentration that yields an adequate level of sedation, while simultaneously ensuring safety for the patient, through an improved predictability of the corresponding MAP and HR responses. Classically, a TCI pump aims to reach and maintain the requested target as fast as possible. However, for dexmedetomidine, that would sometimes require an excessive speed of infusion above 6 mcg/kg/h, yielding an unacceptable risk for acute hypertension at induction. Therefore, the authors advice is to apply the PKPD models only in TCI systems that limit the administration rate of dexmedetomidine to its advised maximum of 6 microg/kg/h, despite the inevitable consequence of a slower onset of effect and a longer duration before the steady state is reached at the requested target. The Hannivoort-Colin PKPD model, including the standardized safety advice for maximum drug administration rate, may therefore ensure optimal effectiveness and safety simultaneously, illustrating the “general purpose” of these combined PD models.

### 4.6. The Morse PK Model for Dexmedetomidine in Children and Adults

Morse et al. aimed to develop a ‘universal’ dexmedetomidine PK model that applies to both children and adults [17]. For this study the authors used data from several existing trials: the Hannivoort study (adults), the Potts study (children), the Cortinez study (obese and non-obese adults), the Rolle study (obese and non-obese adults) and the Talke study (adults) [16,17,34,38,39,40,41]. Interestingly, the Hannivoort PK model performed well in predicting the concentrations in the Potts pediatric study (precision of 22% and bias of 3%) in children > 1 years, presumably a consequence of the allometric scaling used in the Hannivoort model.

Similar to the other global purpose models, some covariates seem to consistently reappear as relevant covariates in the final pharmacokinetic model: the use of fat free mass for clearance, the use of allometric scaling, and the use of a maturation factor based on post-menstrual age (Table 3). One major difference between the Hannivoort and Morse models is the volume of the central compartment V1. In the Hannivoort model, this value is low (1.78 L/70 kg), whereas in the Morse model, this is significantly higher (25.2 L/70 kg). Morse et al. argue that the lower V1 of the Hannivoort model is due to the vasoconstriction induced at higher concentrations and that their estimation of V1 aligns with other models for dexmedetomidine [34,38,42]. However, this does not imply that these values must be correct. The estimation of V1 during PKPD trials is difficult, because the assumption in most PKPD models is that the drug mixes instantly in V1, which is inconsistent with the reality of an intravenous circulation that requires some time to reach homogenous plasma concentrations throughout the blood volume. This is an important reason for overestimating V1 in many studies. In the Hannivoort methodology, a short initial infusion followed by one or more drug samples was performed to bypass this problem, as proposed by Avram et al. [43] Pre-study simulations were done to assess the optimal dosing and timing of samples to estimate V1. Also, Morse et al. did not use the actual data collected from the Hannivoort study, but used simulated datapoints derived from the publication. As the samples at 2 min after the initial short infusion were not publicly available at the time, this information remained unavailable for the Morse model, resulting in a significant loss of data for the estimation of V1.

This apparently technical discussion on the value of V1 might have consequences for clinical safety when applying both models for dexmedetomidine administration. Eleveld et al. published a graph demonstrating this problem in a response letter to the publication of the Morse model [44]. When a plasma concentration is targeted, the administration predictions of the Morse model will consistently yield a significantly higher loading dose compared to the Hannivoort model. An often-advised induction dose of 1 µg kg^−1^ over 10 min results in an infusion rate of 6 µg/kg/h. The low V1 of the Hannivoort model results in a small bolus dose followed by an infusion rate that remains below 6 µg/kg/h, whereas the Morse model would require a larger induction bolus dose, which, if no infusion rate limit is used, could result in undesired hypertension and bradycardia. In a recent follow up simulation study, Morse et al. confirmed that the initial loading dose, delivered by the Morse model, should also be administered over a longer time frame (30 min) to mitigate hemodynamic instability [45]. As not all pump manufacturers realise the need for a limited infusion rate, and not all clinicians are aware that the pumps do not limit infusion rate, for now, the safest model to use would be the Hannivoort model, regardless of whether the Hannivoort V1 or the Morse V1 ends up being the most accurate estimate. Despite mathematical differences between models, the clinical applicability needs to be validated in a prospective study, both for children and adults. Eventually, it is the combined definition of all covariates (not only the volume of V1) that will result in the clinical performance of a PKPD model, which needs further prospective validation for both the Hannivoort and Morse models.

## 5. Conclusions

The new general purpose models have the potential to increase the clinical acceptance of TCI because the models have fewer restrictions for patient selection compared to the currently commercialised models. When applied in a TCI pump, the new models will adapt the doses and infusion rates with similar or even improved accuracy and precision, as well as to a more plausible covariate selection. They avoid mathematical extrapolation problems such as those experienced with the James equation for LBM by using allometric scaling as a rational solution for patients with demographic characteristics at the edges of the demographic spectrum. The main advantage of the new general purpose models is that the anaesthesia practitioners will experience less burden to decide which TCI model represents their patient best.

Although not intended as a general purpose model, the Hannivoort-Colin model for dexmedetomidine may promote new indications and a safer infusion of dexmedetomidine in adult patients. As confirmed in the study by Morse et al., the Hannivoort pharmacokinetic model appears to yield surprisingly accurate predictions of the plasma concentration in children > 1 year of age. Although the Morse model is derived from a larger population and has the potential to have general purpose applicability, some methodological and safety issues should be considered when dexmedetomidine is administered in a target controlled infusion. The first prospective validation studies for the general purpose models of Eleveld (propofol) and Hannivoort (dexmedetomidine) provide promising results and should be considered as incentive to continue research in this direction.

## Figures and Tables

**Table 1 jcm-11-02487-t001:** Demographic ranges of the respective original study populations for developing the PKPD models applied in contemporary commercialized target controlled infusion systems. (non-exhaustive list).

	Population Specific Models	Age-Range (Years)	Weight-Range (kg)	Height-Range (cm)	Number of Subjects	General Purpose Models	Age-Range (Years)	Weight-Range (kg)	Height-Range (cm)	Number of Subjects
**PROPOFOL**										
	Marsh [4]	2–17	12–54	NR	37	Eleveld [5]	0–88	0.68–160		1033
	Schnider [6,7]	26–81	44–123	155–196	24					
	Paedfusor [8]	2–13	NR	NR	NR					
	Cortinez (Obese) [9]	21–53	85–141	148–178	19					
	Cortinez (Children) [10]	0–3	5.2–11.4	57–79	41					
	Kataria [11]	3–11	15–61	NR	53					
**REMIFENTANIL**										
	Minto [12]	20–85	48–108 ~	156–192 ~	65	Eleveld [13]	0–85	2.5–106	49 -193	131
						Kim-Obara-Egan [14]	20–85	45–215	150–196	229
**SUFENTANIL**										
	Gepts [15]	14–68	47–94	154–182	23	NR				
**DEX-MEDETOMIDINE**										
	Hannivoort [16]	18–72	51–110	NR	18	Morse [17]	0–71	3.1–152	55–180 *	202

NR = Not reported, * approximate values derived from Figure 2.1 in [17], ~ approximate values derived from Figure 1 in [12].

**Table 2 jcm-11-02487-t002:** Different equations to define the size of patients.

	Male	Female
Lean body mass (LBM) as proposed by **James** [24] (Size descriptor in the Schnider model for propofol [7] and the Minto model for remifentanil [12]	LBM = 1.1 × w − 128 × w^2^/h^2^	LBM = 1.07 × w − 148 × w^t^/h^2^
Fat free mass (FFM) as proposed by **Janmahastian et al.** (Size descriptor in the Kim-Obara-Egan model for remifentanil [14]. Applicable for adults only)	$\mathrm{FFM}=\frac{9.27\times 10^{3}\times w}{6.68\times 10^{3}+216\times \mathrm{BMI}}$	$\mathrm{FFM}=\frac{9.27\times 10^{3}\times w}{8.78\times 10^{3}+244\times \mathrm{BMI}}$
Fat free mass (FFM) as proposed by **Al Sallami et al.** (Size descriptor in the Eleveld models for propofol [5] and remifentanil [13]. Applicable for adults and children)	$\mathrm{FFM}=\left[0.88+\frac{1-0.88}{1+{\left(\frac{\mathrm{AGE}}{13.4}\right)}^{-12.7}}\right]\;\times$ $\left[\frac{9270\times w}{6680+216\times \mathrm{BMI}}\right]$	$\mathrm{FFM}=\left[1.11+\frac{1-1.11}{1+{\left(\frac{\mathrm{AGE}}{7.1}\right)}^{-1.1}}\right]\;\times$ $\left[\frac{9270\times w}{8780+244\times \mathrm{BMI}}\right]$
Normal fat mass (NFM) as proposed by **Morse et al.** [17]	FAT=w−FFM NFM=FFM+Ffat×FAT	FAT=w−FFM NFM=FFM+Ffat×FAT

w = total body weight in kg, h = height in cm, BMI = body mass index, AGE = in years, Ffat = factor for normal fat mass, estimated with NONMEM or fixed to 0.

**Table 3 jcm-11-02487-t003:** The four general purpose pharmacokinetic models.

**Propofol: Eleveld PK-PD model** [5]
Fsize = weight/70
Fage(x) = exp(x *(age − 35))
Fsigmoid(x, e50, gamma) = x **gamma/(x **gamma + e50 **gamma)
Fcentral = Fsigmoid(weight, 33.6, 1)
Fopiates(x) = absence: 1, presence: exp(x *age)
FmatCL = Fsigmoid(post-menstrual age, 42.3 weeks, 9.06)
FsexCL = male: 1.79, female: 2.10
FmatQ3 = Fsigmoid(age + 40 weeks, 68.3 weeks, 1)
V1 = 6.28 *(Fcentral(weight)/Fcentral_ref_)
V2 = 25.5 *Fsize *Fage(−0.0156)
V3 = 273 *(FFM (Al-Sallami)/FFMref) *Fopiates(−0.0138)
CL1 = Fsex *Fsize **0.75 *(FmatCL/FmatCL_ref_) *Fopiates(−0.00286)
CL2 = 1.75 *(V2/V2ref) **0.75 *(1 + 1.3 *(1 – FmatQ3))
CL3 = 1.11 *(V3/V3ref) **0.75 *(FmatQ3/FmatQ3_ref_)
ke0 = 0.146 *Fsize **−0.25
E50 = 3.08 *Fage(−0.00635)
**Remifentanil: Eleveld PK-PD model** [13]
Fsize = FFM(Al-Sallami)/FFM_ref_
Fage(x) = exp(x *(age − 35))
Fsigmoid(x, e50, gamma) = x **gamma/(x **gamma + e50 **gamma)
Fmat(weight) = Fsigmoid(weight, 2.88, 2)
Fsex = male: 1, female: 1 + 0.47 *Fsigmoid(age, 12, 6) *(1 − Fsigmoid(age, 45, 6))
V1 = 5.81 *Fsize *Fage(−0.00554)
V2 = 8.82 *Fsize *Fage(−0.00327) *Fsex
V3 = 5.03 *Fsize *Fage(−0.0315) *exp(−0.0260 *(weight − 70))
CL1 = 2.58 *Fsize **0.75 *(Fmat/Fmat_ref_) *Fsex *Fage(−0.00327)
CL2 = 1.72 *(V2/8.82) **0.75 *Fage(−0.00554) *Fsex
CL3 = 0.124 *(V3/5.03) **0.75 *Fage(−0.00554)
ke0 = 1.09 *Fage(−0.0289)
**Remifentanil: Kim-Obara-Egan model** [27]
V1 = 4.76 *(weight/74.5) **0.658
V2 = 8.4 *(FFM (Janmahasatian)/52.3) **0.573 – 0.0936 *(age − 37)
V3 = 4 – 0.0477 *(age − 37)
CL1 = 2.77 *(weight/74.5) **0.336 – 0.0149 *(age − 37)
CL2 = 1.94 – 0.0280 *(age − 37)
CL3 = 0.197
**Dexmedetomidine: Morse model** [17]
V1 = 25.2 L/70 kg NFM
V2 = 34.4 L/70 Kg NFM
V3 = 65.4 L/70 kg NFM
CL1 = 0.897 L/min/70 kg FFM(Al-Sallami)
CL2 = 1.68 L/min/70 kg FFMCL3 = 0.62 L/min/70 kg FFM

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
