# Peer review of "General Purpose Pharmacokinetic-Pharmacodynamic Models for Target-Controlled Infusion of Anaesthetic Drugs: A Narrative Review"

_jcm, 2022, doi:10.3390/jcm11092487_

Round 1

Reviewer 1 Report

This is a very elegant review of the topic that will be of great interest to the anaesthesia community and likely to be cited. There is no doubt that the Eleveld model is a major innovation in TCI and has been well described here.

Minor comments: the use of BIS as a surrogate endpoint of consciousness has a number of limitations given the variability between subjects and various factors that can affect this number. Perhaps a brief discussion of this would be helpful.

The authors may also wish to include more information of the time to peak effect as this has an important bearing on the predicted effect site concentration of the different models (particularly propofol).

Author Response

Reviewer 1

This is a very elegant review of the topic that will be of great interest to the anaesthesia community and likely to be cited. There is no doubt that the Eleveld model is a major innovation in TCI and has been well described here.

A: We thank the reviewer for his positive impression. The second reviewer did point out some omissions which have been corrected in this new revised  version. We hope you remain equally positive for this improved reviewed version.

Minor comments: the use of BIS as a surrogate endpoint of consciousness has a number of limitations given the variability between subjects and various factors that can affect this number. Perhaps a brief discussion of this would be helpful.

A: We mention the use of different EEG derived indices as one of the limitations of the contemporary TCI models. We do realize that the EEG measurements remain a surrogate quantification of hypnotic effect. However, as pharmacodynamic measures of individual sensitivity for anaesthetic drugs, the EEG monitors appear to do a fine job. The discussion on the value of the EEG indices is rather complex and could complicate the review with excessive technical details. We do hope you accept that we did not elaborate further on the position of EEG monitoring  as this might be somewhat outside the scope of our review goal.

The authors may also wish to include more information of the time to peak effect as this has an important bearing on the predicted effect site concentration of the different models (particularly propofol).

 A: We thank the reviewer for this constructive comment. Also for this topic, we do realize the implications of using a modelled or rather a TTPE derived effect-site concentration. We did add a limitation of the models related to the TTPE method for defining ke0. However, In order not to exceed the amount of technical details, we prefer not to add additional technical discussions on the difference between the modelled approach versus the TTPE method. Our review is intended to be appealing for the novice in the domain of modeling,  who are confronted with a representative of a company who aims to sell a TCI pump with new models. We think the TTPE discussion is not central in that decision. We hope this is an acceptable solution for reviewer 1.

Reviewer 2 Report

General purpose pharmacokinetic-dynamic models for target- controlled infusion of anaesthetic drugs. A narrative review.

The authors present a narrative review concerning pharmacokinetic pharmacodynamic models for intravenous drugs used in target-controlled infusion (TCI) systems.

Parameter sets derived from a broad population (e.g., children to adults) are reviewed based on the premise that there is a lack of generalizability in models currently programmed into TCI pumps.

Models for commonly used drugs (propofol, remifentanil, dexmedetomidine) are summarized. The authors demonstrate the utility of covariate modelling to account for pharmacokinetic variability between individuals.

The primary differences between children and adults concern size and age. Theory based allometry and maturation of clearance with age can be used to describe these differences and are fundamental to general purpose models. I believe these core concepts need to be clearer throughout the manuscript.

The manuscript requires editing for grammar, clarity and flow. The depth and breadth of information can be extended in some areas. Please see my specific comments below. 

Title

Please change pharmacokinetic-dynamic to pharmacokinetic pharmacodynamic.

Summary

Lines 27-29 Please link this point back to drug concentration. While the pharmacokinetic parameters (clearance and volume) are used to calculate drug dose the system relies on a user-selected drug concentration.

It would also be useful to mention differences between plasma and effect-site targeting. The equilibration rate constant for some drugs is not available hence some pumps can only be used in target plasma concentration mode. Others enable target effect-site concentration mode, but the T1/2keo is estimated by means other than PKPD modelling

Line 32 Please clarify why these populations are more “challenging” e.g. due to the maturation of clearance in neonates and infants affecting drug dose, as does decline in organ function in the elderly.

Line 33 Delete the word “behaviour” and the words “as such”.

Line 39 Please explain what you mean by “dedicated covariate selection”. Covariates should be implemented in a model based on their biological plausibility. Age and weight are often the best place to start and typically describe most of the variability between individuals. Pharmacogenetics are rarely implemented. Organ dysfunction is rarely used.

Introduction

Lines 52-53 Please reframe the sentence “For most anaesthetic drugs, a multi-compartmental model, that combines connected volumes of distribution and parameters for clearance is used as the initial structural pharmacokinetic model”.
The two main pharmacokinetic parameters are clearance and volume of distribution. This phrasing is confusing i.e. parameters for clearance. Clearance is the parameter.

Line 55 The sentence “By adding relevant demographic and/or physiologic covariates into the equations for the volumes of distribution and/or clearances, the prediction accuracy of the final model may improve even more.” can be improved. Incorporating covariates into a pharmacokinetic model accounts for predictable sources of variability.  Please frame as such rather than the vague assertion of “improving the final model even more”. The implication is usually that covariate use reduces between subject variability.

Lines 64-66 This point could be strengthened if linked back to the pharmacokinetic parameters clearance and volume of distribution. Volume and clearance are used to calculate loading dose and maintenance dose at steady-state, respectively. Population parameters are incorporated into the pump, covariates (e.g. weight, age) are used to determine an induvial parameter when random effects are accounted for.

Line 92 Allometry would be more correctly described as a way to incorporate covariates relating to size which allows for extrapolation between populations. There are many studies which describe allometry in anaesthesia, its applicability in children and the extension to body composition (Anderson and Holford 2008, Holford and Anderson 2017, Gonzalez-Sales, Holford et al. 2022).

Table 1

It would be useful to include which covariates were retained in the final model for these parameter sets.

There are many models missing from this table and the associated text. I think it is relevant to extend this to other anaesthetic and analgesic drugs for which many models are available. Many of these models also describe the combined influence of anaesthetic agents on measures of effect which is more interesting and useful. I have included some examples below:

Ketamine  pharmacokinetics in children (Herd, Anderson et al. 2007).

Compartmental methods have also been employed for the analysis of sevoflurane pharmacokinetics: (Cortinez and Anderson 2018)

Interaction models have described the combined effect of remifentanil and propofol on bispectral index (Bouillon, Bruhn et al. 2004, Fuentes, Cortinez et al. 2018)

There are even more models using size and allometry for perioperative medication ( e.g., acetaminophen (Anderson, Holford et al. 1997, Anderson, Woollard et al. 2000)) and physiological functions (e.g., renal function (Rhodin, Anderson et al. 2009). Use of these models is extensive.(Holford, Heo et al. 2013)

Comparing the performance of multiple PKPD models

These concepts were introduced prior to Varvel by Sheiner and Beal (Sheiner, Beal et al. 1981)

Limitations of contemporary TCI models

Lines 120-121 and elsewhere The sentence: “For some drugs, such as propofol, multiple PKPD models are available, revealing significantly different parameterizations of the structural model and added covariates.” excessively uses commas and is clumsy to read. Please limit the number of introductory clauses used in this manuscript (e.g. For some drugs,..)

Line 120 Please clarify “significantly different parameterizations  of the structural model”. The word significantly should be used in context of statistical significance with some sort of evidence.
Are you referring to parameter estimates? Is the word simply an adjective with no statistical meaning?

Different parametrizations would indicate to me using differential equations with rate constants and amounts rather than concentrations and clearances.

Table 2

The authors should include the size descriptor normal fat mass. Normal fat mass is an extension of fat free mass and describes contributions from both fat and fat-free mass. This contribution is described by the parameter Ffat which is specific for each drug and clearance and volume of distribution, 

Normal fat mass can be scaled using theory based allometry (Holford and Anderson 2017). It just so happens that Ffat=0 for remifentanil and Ffat=1 for propofol. This should not imply that FFM and TBM are the only size models for consideration (Anderson and Holford 2017).

Dexmedetomidine (Morse, Cortinez et al. 2020), paracetamol (Allegaert, Olkkola et al. 2014) and busulfan are examples where normal fat mass has proven to be a useful size descriptor (McCune, Bemer et al. 2014).

The open TCI initiative

Line 164 Perhaps include that propofol and remifentanil are the main (only?) datasets available on this site.

Please add that these are deidentified data.

Data sharing is also common between research centres. The ethics and etiquette of data sharing in pharmacokinetic studies has been discussed widely (Anderson and Merry 2009).

Line 180 Please change demographic parameter to demographic variable.

Line 183 Please note that the Al-Sallami formulae for FFM is applicable to individuals down to 3 years.(Al-Sallami, Goulding et al. 2015)

Table 3

This is simply a collection of equations rather than a table. The presentation of equations this way is probably of little value to the typical reader. I suggest the authors reformat this table or combine it with Table 1 to demonstrate clearly which covariates are included in each final model.

The dexmedetomidine Morse model is a three-compartment structural model. The intercompartmental clearance for the third (Q3) compartment is missing.

The best size descriptor for dexmedetomidine volume of distribution was normal fat mass not fat-free mass as the authors state here. Dexmedetomidine FfatV 0.293 (Morse, Cortinez et al. 2020).

Line 239 Delete the after 3/4

Line 251 (and elsewhere) Reframe sentence to avoid statements like “he/she”. Referring to the individual as an anaesthetic practitioner is more appropriate.

The Eleveld PKPD model for remifentanil

Please check this section for grammar and flow. It is difficult to read with the haphazard use of commas breaking up sentences.

Line 284 Replace “has been published” to have been published.

Lines 285-286 Avoid vague statements such as “a rather small range of body mass indices” quantify the range instead.

Line 293 Which body size descriptor?

Line 299 What is considered to be favourable MDPE and MDAPE?

Line 300 Delete the word noteworthy

The Kim-Obara-Egan PK model for remifentanil

Lines 317-317 This introductory sentence about the parallel development of models is unnecessary.

Line 332 Please change “power factor” to allometric exponent.

The Hannivoort-Colin PKPD model for dexmedetomidine

Line 340-341 Please clarify the point about the dexmedetomidine biphasic adverse effect profile. These adverse effects are driven by drug concentration. At concentrations exceeding 2 ng/mL hypertension predominates while at lower concentrations dexmedetomine induces hypotension. This is attributed to direct vasoconstrictor effect and delayed central vasodilator effect. (Potts, Anderson et al. 2010, Talke and Anderson 2018)

Line 344-355 Dexmedetomidine infusion of 0.5-0.1 mcg/kg over 10 minutes is to prevent these concentration dependent adverse effects you mention above rather than anything to do with reaching steady state. A large dose could be administered to reach steady state rapidly but this would be at the cost of haemodynamic stability (Morse, Cortinez et al. 2021).

Delay in onset is quantified by the equilibration half-time (around 3 minutes for sedation) (Li, Yuen et al. 2018)

Line 357 Please clarify what you mean by blind spots in the data.

Line 358 Please replace the word gender with sex. Sex is genotype. Gender is phenotype.

Line 420 Please replace the word efficacy with effectiveness. Efficacy here really refers to maximum effect on an Emax curve

The Morse PK model for dexmedetomidine in children and adults

Line 434 As mentioned in Table 3 the Morse model found fat free mass to be the best size descriptor for clearance but normal fat mass superior to both fat free mass and total body weight for volume of distribution.

It is worth noting that the central volume of distribution 25 L/70 kg obtained by Morse aligns with other published studies in children and adults (Dyck, Maze et al. 1993, Potts, Warman et al. 2008, Talke and Anderson 2018). They could offer no explanation for why the Hannivoort model V1 is at odds with every other published paper. They simply suggested maybe vasoconstriction contributed. This is obviously a contentious point, but probably of low importance. Have the authors reviewed their model using a larger V1. How much does this alter the OBJ?

It is true that a smaller V1 will yield a smaller dose potentially mitigating the concentration-dependent adverse effects that are associated with a rapid bolus producing high concentrations. However, conversely a small estimate of V1 may not attain a steady state concentration rapidly due to a small dose if V1 bigger.

The authors should recognize that Morse and colleagues subsequently published a paper of pharmacokinetic concepts for dexmedetomidine target controlled infusion pumps. Here they demonstrate using their pharmacokinetic model and a model for heart rate and blood pressure that haemodynamic instability can be mitigated with an infusion over 15 minutes, despite a higher dose being required to attain a target effect of sedation compared with a  bolus dose (Morse, Cortinez et al. 2021) .

It is also important to realise that with an infusion over a given time, volume is not the sole parameter determining dose. Intercompartment clearances and other compartment volumes are important. 

I agree with the authors that an infusion rate limit on TCI pumps is important for patient safety.

Author Response

Reviewer 2

General purpose pharmacokinetic-dynamic models for target- controlled infusion of anaesthetic drugs. A narrative review.

The authors present a narrative review concerning pharmacokinetic pharmacodynamic models for intravenous drugs used in target-controlled infusion (TCI) systems.

Parameter sets derived from a broad population (e.g., children to adults) are reviewed based on the premise that there is a lack of generalizability in models currently programmed into TCI pumps.

Models for commonly used drugs (propofol, remifentanil, dexmedetomidine) are summarized. The authors demonstrate the utility of covariate modelling to account for pharmacokinetic variability between individuals.

The primary differences between children and adults concern size and age. Theory based allometry and maturation of clearance with age can be used to describe these differences and are fundamental to general purpose models. I believe these core concepts need to be clearer throughout the manuscript.

The manuscript requires editing for grammar, clarity and flow. The depth and breadth of information can be extended in some areas. Please see my specific comments below. 

A: We thank the reviewer for the thorough evaluation and constructive comments.

Title

Please change pharmacokinetic-dynamic to pharmacokinetic pharmacodynamic.

A: We adjusted this as requested.

Summary

Lines 27-29 Please link this point back to drug concentration. While the pharmacokinetic parameters (clearance and volume) are used to calculate drug dose the system relies on a user-selected drug concentration.

A: We adjusted this as requested.

It would also be useful to mention differences between plasma and effect-site targeting. The equilibration rate constant for some drugs is not available hence some pumps can only be used in target plasma concentration mode. Others enable target effect-site concentration mode, but the T1/2keo is estimated by means other than PKPD modelling

A: We now added a remark in the summary related to  the limits of  models that are restricted to plasma controlled TCI versus effect-site controlled TCI and shortly mention the use of alternative methods to define the equilibration rate constant of the effect-site. However, we chose to limit the degree of detail in the summary.  Rather the goal of the summary is to trigger the reader to continue reading to find more details and nuance in the manuscript.

Line 32 Please clarify why these populations are more “challenging” e.g. due to the maturation of clearance in neonates and infants affecting drug dose, as does decline in organ function in the elderly.

A: We changed the sentence to: … , often excluding the more challenging populations, such as children, obese or elderly patients, of whom the body composition or elimination mechanisms may be  structurally different compared to the  lean adult patient population. Separate PKPD models have been developed for some of these subcategories, but the availability of multiple PKPD models for a single drug  increases the risk for invalid model selection by the user.

Line 33 Delete the word “behaviour” and the words “as such”.

A: We adjusted this as requested.

Line 39 Please explain what you mean by “dedicated covariate selection”. Covariates should be implemented in a model based on their biological plausibility. Age and weight are often the best place to start and typically describe most of the variability between individuals. Pharmacogenetics are rarely implemented. Organ dysfunction is rarely used.

A: Thank you for this suggestion:  we decided to rephrase this paragraph to improve clarity.

We wrote: “Advances in computing and the emergence of globally collected databases has allowed the development of new “general purpose” PKPD models. These take on the challenging task of identifying the relationships between patient covariates (age, weight, sex, etc) and the volumes and clearances of multi-compartmental  pharmacokinetic models applicable across broad populations from neonates to the elderly, from the underweight to the obese. These models address the issues of allometric scaling of body weight and size, body composition, sex differences, changes with advanced age, and for young children, changes with maturation and growth. General purpose models for propofol, remifentanil and dexmedetomidine have appeared and these greatly reduce the risk of invalid model selection.“

Introduction

Lines 52-53 Please reframe the sentence “For most anaesthetic drugs, a multi-compartmental model, that combines connected volumes of distribution and parameters for clearance is used as the initial structural pharmacokinetic model”.
The two main pharmacokinetic parameters are clearance and volume of distribution. This phrasing is confusing i.e. parameters for clearance. Clearance is the parameter.

A: we rephrased the sentence to:  For most anaesthetic drugs, the initial structural pharmacokinetic model is defined as a multi-compartmental model with a specific volume of distribution and clearance for each compartment.

Line 55 The sentence “By adding relevant demographic and/or physiologic covariates into the equations for the volumes of distribution and/or clearances, the prediction accuracy of the final model may improve even more.” can be improved. Incorporating covariates into a pharmacokinetic model accounts for predictable sources of variability.  Please frame as such rather than the vague assertion of “improving the final model even more”. The implication is usually that covariate use reduces between subject variability.

A: We thank the reviewer for the valued suggestions for improvement. We changed this section to: Incorporating relevant demographic and/or physiologic covariates into the equations for  volumes of distribution or rate constants or clearance, does account for predictable sources of biological variability. This implies that a model with accurate covariate selection often will reduces the between subject variability in the model predictions of the plasma concentration compared to a model without -or with suboptimal- selection of demographic covariates.

Lines 64-66 This point could be strengthened if linked back to the pharmacokinetic parameters clearance and volume of distribution. Volume and clearance are used to calculate loading dose and maintenance dose at steady-state, respectively. Population parameters are incorporated into the pump, covariates (e.g. weight, age) are used to determine an induvial parameter when random effects are accounted for.

A: We thank the reviewer for these suggestions. We altered this section to: “Rather than calculating the bolus dose or infusion rate  in linear weight-adjusted fashion, the volumes of distribution and clearance rates, as (non linearly) defined by the PKPD model, will determine the initial bolus dose and subsequent adaptations in the drug administration rate,  to reach and maintain a steady state in either the plasma- or the effect-site concentration. The input of individual demographic variables in the models’  equations further adds an individual factor to the drug administration, after other random sources of biological variability are accounted for.”

We also rephrased the section on the need for adjusting the target according to the observed effect, despite the individual demographic data input in the models. Line 76 to 82 (page 4).

Line 92 Allometry would be more correctly described as a way to incorporate covariates relating to size which allows for extrapolation between populations. There are many studies which describe allometry in anaesthesia, its applicability in children and the extension to body composition (Anderson and Holford 2008, Holford and Anderson 2017, Gonzalez-Sales, Holford et al. 2022).

A: We thank the reviewer for this suggestion. We rephrased the definition of allometry to: Allometry aims to describe the mathematical relationship between biologic processes across a diversity of sizes.(8) This may appear a valuable theory to define pharmacokinetic characteristics of subjects at the extremes of the demographic spectrum or to allow for extrapolation between populations.

Table 1

It would be useful to include which covariates were retained in the final model for these parameter sets.

A: We like to point out that the complete model parameter sets (pharmacokinetics) are shown in table 3 of the manuscript as requested by the reviewer.

There are many models missing from this table and the associated text. I think it is relevant to extend this to other anaesthetic and analgesic drugs for which many models are available. Many of these models also describe the combined influence of anaesthetic agents on measures of effect which is more interesting and useful. I have included some examples below:

Ketamine  pharmacokinetics in children (Herd, Anderson et al. 2007).

Compartmental methods have also been employed for the analysis of sevoflurane pharmacokinetics: (Cortinez and Anderson 2018)

Interaction models have described the combined effect of remifentanil and propofol on bispectral index (Bouillon, Bruhn et al. 2004, Fuentes, Cortinez et al. 2018)

There are even more models using size and allometry for perioperative medication ( e.g., acetaminophen (Anderson, Holford et al. 1997, Anderson, Woollard et al. 2000)) and physiological functions (e.g., renal function (Rhodin, Anderson et al. 2009). Use of these models is extensive.(Holford, Heo et al. 2013)

A: We thank the reviewer for this suggestion. We agree that this manuscript is far from complete if it was the intention to describe all published PKPD models that apply allometry or focus on “challenging” populations or are derived from databases with a wide range in demographic characteristics of the studied population. However, the goal of our review paper was to target an audience of clinicians who like to start using the general purpose models in clinical practice. We therefore focused -to the best of our knowledge-  exclusively on the PKPD models that are currently commercially available in TCI pumps (or will be commercialized soon). We know that in literature, many more PKPD models are available, including some that also apply allometry into their final parameter sets. Some of these models are also used prospectively for clinical research. However, in view of the targeted audience, we found it more useful to limit the extend of our topics in this article to the clinically available models. We did alter the title of Table 1 to make this clearer:

“Table 1: A non-exhaustive list of demographic ranges of the respective original study populations for developing the  PKPD models applied in contemporary commercialized target controlled infusion systems.”

We also conclude our introduction with: “The focus of the review is restricted to models that are currently applied  in a commercially distributed TCI pump, or for which -to the best of our knowledge- a commercial applicability will be available soon. However, as the commercialization of technology is a continuous and  non-exhaustive process, the authors  do not claim completeness. However, we found that the models included in this review provide a sufficient illustrative framework to point out innovations and limitations of  the “general purpose” PKPD technology.”

 Comparing the performance of multiple PKPD models

These concepts were introduced prior to Varvel by Sheiner and Beal (Sheiner, Beal et al. 1981)

A: Thank you for pointing this out. We added this in the reference list and refer to the work of Sheiner and Beal in the manuscript (line 124)

Limitations of contemporary TCI models

Lines 120-121 and elsewhere The sentence: “For some drugs, such as propofol, multiple PKPD models are available, revealing significantly different parameterizations of the structural model and added covariates.” excessively uses commas and is clumsy to read. Please limit the number of introductory clauses used in this manuscript (e.g. For some drugs,..)

Line 120 Please clarify “significantly different parameterizations  of the structural model”. The word significantly should be used in context of statistical significance with some sort of evidence.
Are you referring to parameter estimates? Is the word simply an adjective with no statistical meaning?

Different parametrizations would indicate to me using differential equations with rate constants and amounts rather than concentrations and clearances.

A: We rephrased the sentence to: Multiple PKPD models have been published to describe the PKPD of the same drug, but they often differ in the final selection of the structural multicompartmental model and added covariates.

Table 2                    

The authors should include the size descriptor normal fat mass. Normal fat mass is an extension of fat free mass and describes contributions from both fat and fat-free mass. This contribution is described by the parameter Ffat which is specific for each drug and clearance and volume of distribution, 

Normal fat mass can be scaled using theory based allometry (Holford and Anderson 2017). It just so happens that Ffat=0 for remifentanil and Ffat=1 for propofol. This should not imply that FFM and TBM are the only size models for consideration (Anderson and Holford 2017).

Dexmedetomidine (Morse, Cortinez et al. 2020), paracetamol (Allegaert, Olkkola et al. 2014) and busulfan are examples where normal fat mass has proven to be a useful size descriptor (McCune, Bemer et al. 2014).

A: We thank the reviewer for recognizing this missing information. We indeed added the equations for Normal Fat mass in Table 2 as it was used in the Morse model. We also adjusted the error (changing FFM to NFM) in table 3 that shows all parameters of the discussed models in this review.

The open TCI initiative

Line 164 Perhaps include that propofol and remifentanil are the main (only?) datasets available on this site.

Please add that these are deidentified data.

Data sharing is also common between research centres. The ethics and etiquette of data sharing in pharmacokinetic studies has been discussed widely (Anderson and Merry 2009).

Line 180 Please change demographic parameter to demographic variable.

A: We added  information on the three previous remarks to the paragraph on the open TCI initiative. Line 182-184

Line 183 Please note that the Al-Sallami formulae for FFM is applicable to individuals down to 3 years.(Al-Sallami, Goulding et al. 2015)

A: Thank you for this valuable comment. We added this information in the manuscript.

Table 3

This is simply a collection of equations rather than a table. The presentation of equations this way is probably of little value to the typical reader. I suggest the authors reformat this table or combine it with Table 1 to demonstrate clearly which covariates are included in each final model.

A: We thank the reviewer for this suggestion. However, we deliberately preferred not to mix table 1 and 3 in order to make a distinction between the basic knowledge every TCI user needs (For which demographic range is a PKPD model applicable?) and the information that might be more of interest to the expert reader (How are the covariates integrated into the multi-compartmental model?) We also prefer to edit the formulas in a Table format, as this allows to include the information to remain in the main manuscript, rather than as a separate online appendix. By combining all information in one single table, we fear excessive complexity which may cause a sense of intimidation for the novice in the domain of PKPD modeling. We hope this editorial choice is acceptable for the editor and reviewer.

The dexmedetomidine Morse model is a three-compartment structural model. The intercompartmental clearance for the third (Q3) compartment is missing.

The best size descriptor for dexmedetomidine volume of distribution was normal fat mass not fat-free mass as the authors state here. Dexmedetomidine FfatV 0.293 (Morse, Cortinez et al. 2020).

A: The missing Q3 was an accidental omission. Also the correction regarding NFM is now changed in table 3

Line 239 Delete the after ¾

A: Done

Line 251 (and elsewhere) Reframe sentence to avoid statements like “he/she”. Referring to the individual as an anaesthetic practitioner is more appropriate.

A: Agreed. We have adjusted this as requested.

The Eleveld PKPD model for remifentanil

Please check this section for grammar and flow. It is difficult to read with the haphazard use of commas breaking up sentences.

A: We restructured some sentences to improve the flow of the paragraph for the reader.

Line 284 Replace “has been published” to have been published.

A: Done

Lines 285-286 Avoid vague statements such as “a rather small range of body mass indices” quantify the range instead.

A: Unfortunately, the BMI range was not reported in the Minto paper, and the LBM was only published as a figure. As we were not able to describe the BMI range precisely, we used a subjective description. We agree that this is a suboptimal solution. So we now changed the sentence to: The Minto model is based on a dataset of 65 non obese adults, aged between 20 to 77 years.

Line 293 Which body size descriptor?

A: The exact size descriptor used is called FSize, which is the Fat free mass scaled to a reference Fat free mass. In order not to complicate the sentence we did not specify that. For clarity, we will change the statement to clarify that the size descriptor is related to FFM.

Line 299 What is considered to be favourable MDPE and MDAPE?

A: We removed the word favourable and added the minimal and maximal values.

Line 300 Delete the word noteworthy

A:Done

The Kim-Obara-Egan PK model for remifentanil

Lines 317-317 This introductory sentence about the parallel development of models is unnecessary.

A: we removed the sentence.

Line 332 Please change “power factor” to allometric exponent.

A: Done

The Hannivoort-Colin PKPD model for dexmedetomidine

Line 340-341 Please clarify the point about the dexmedetomidine biphasic adverse effect profile. These adverse effects are driven by drug concentration. At concentrations exceeding 2 ng/mL hypertension predominates while at lower concentrations dexmedetomine induces hypotension. This is attributed to direct vasoconstrictor effect and delayed central vasodilator effect. (Potts, Anderson et al. 2010, Talke and Anderson 2018)

A: We restructured the paragraph to explain the complexity of the dexmedetomidine hemodynamic effects. (We prefer not to call them adverse effects, as they may be a desired goal)  We agree with the proposed mechanisms on how dexmedetomidine evokes a biphasic effect on blood pressure provided that the effects are derived from pharmacological steady-state conditions (after a equilibration time has been maintained). However, this biphasic  concentration effect relationship as predicted by the Hannivoort model can be disrupted by an excessive speed of drug administration. The speed of drug administration is not included as a covariate in the model, but it does play a role in safety, as a fast injection may cause an initial hypertension (evoked by immediate effects on the vascular smooth muscles) only to be smoothed at later stage by the centrally mediated hypotensive effects. Therefore, a slow infusion speed, allows at one hand to minimize the initial peripherally evoked  vasoconstriction and secondly does allow sufficient time to equilibrate the drug at the effect-site  to evoke the centrally mediated hypotensive effect. The latter appears to be most pronounced at lower drug concentrations (below 2ng/ml) , and becomes overruled by increasing peripheral vasoconstriction at higher concentrations above 2ng/ml.

The text has therefore been changed to:

Dexmedetomidine is an alpha-2 adrenoreceptor agonist effective on both presynaptic and postsynaptic receptors, which decreases the sympathetic transmission. It has sedative and analgesic properties,  induces bradycardia with increasing doses and has a complex biphasic effect on blood pressure. At low drug concentrations below 2ng/ml, dexmedetomidine evokes hypotension by activating receptors in the central nervous system. Higher concentrations evoke a progressively intensifying  hypertension through the activation of receptors in the peripheral vascular smooth muscles. However, this biphasic concentration-effect relationship is only valid when the drug is administered in a sufficiently slow infusion speed (below 6 microg/kg/h). Higher administration rates may disrupt this relationship because a faster onset of the peripheral vasoconstrictor effect, may cause significant hypertension and bradycardia, before a slower onset of the centrally mediated hypotensive effect can counter the vasoconstriction.

Line 344-355 Dexmedetomidine infusion of 0.5-0.1 mcg/kg over 10 minutes is to prevent these concentration dependent adverse effects you mention above rather than anything to do with reaching steady state. A large dose could be administered to reach steady state rapidly but this would be at the cost of haemodynamic stability (Morse, Cortinez et al. 2021).

Delay in onset is quantified by the equilibration half-time (around 3 minutes for sedation) (Li, Yuen et al. 2018)

A: We agree and will adapt these inaccuracies in the restructured paragraph.

Line 357 Please clarify what you mean by blind spots in the data.

A: By blind spots in the data we refer to situations where  inclusions only focus on a small range of ages, or for instance when only women of  a younger age are selected in combination with mainly men of an older age. However, we agree that this part of the sentence was confusing and removed this part of the sentence.

Line 358 Please replace the word gender with sex. Sex is genotype. Gender is phenotype.

A: Thank you for this correct comment, we have changed this.

Line 420 Please replace the word efficacy with effectiveness. Efficacy here really refers to maximum effect on an Emax curve

A: we changed the text as suggested.

The Morse PK model for dexmedetomidine in children and adults

Line 434 As mentioned in Table 3 the Morse model found fat free mass to be the best size descriptor for clearance but normal fat mass superior to both fat free mass and total body weight for volume of distribution.

A: NFM is added as a covariate, also table 3 has been corrected..

It is worth noting that the central volume of distribution 25 L/70 kg obtained by Morse aligns with other published studies in children and adults (Dyck, Maze et al. 1993, Potts, Warman et al. 2008, Talke and Anderson 2018). They could offer no explanation for why the Hannivoort model V1 is at odds with every other published paper. They simply suggested maybe vasoconstriction contributed. This is obviously a contentious point, but probably of low importance. Have the authors reviewed their model using a larger V1. How much does this alter the OBJ?

A: The estimation of V1 during PKPD trials is difficult, because one has to contend with front-end kinetics. The assumption in most PKPD models  is that the drug mixes instantly in V1, which is inconsistent with reality. Thus sampling too soon results in too low concentrations because the drug hasn’t mixed in the blood compartment, and often overestimates V1. Sampling too late causes distribution to peripheral compartments to influence the estimation of V1. One possible way to bypass this problem, is to perform a short initial infusion followed by one or more drug samples. This is described by Avram et al. (Anesthesiology. 2003 Nov;99(5):1078-86). This was done for the Hannivoort-Colin model, with pre-study simulations done to assess the dosing and timing of samples to best estimate V1. The fact that other studies all describe higher values of V1 does not mean that those values are correct, as the study methods often cause V1 to be overestimated, and the above explanation and others are described in the comment by Eleveld et al.

The current value of V1 is what led to the lowest objective function in the Hannivoort analysis. Other modeling attempts with larger estimated V1 result in a higher objective function. Despite mathematical differences between models, the clinical applicability needs to be validated in prospective study. Changing only a single covariate (e.g. V1) to try to improve the model, may appear biologically plausible, but eventually, it is the combined definition of all covariates that will result in the clinical performance of a PKPD model. In the clinicians perspective, despite the definition of V1, almost all current PKPD models have the worst performance in the first couple of minutes, which does indicate the relative (un)importance of the discussion who is right and who is wrong on the V1 volume of distribution. It is rather the clinical performance that will decide on the matter which model is most useful.

It is true that a smaller V1 will yield a smaller dose potentially mitigating the concentration-dependent adverse effects that are associated with a rapid bolus producing high concentrations. However, conversely a small estimate of V1 may not attain a steady state concentration rapidly due to a small dose if V1 bigger.

A: This may be true if one assumes the small V1 is incorrect. But as explained above, the fact that V1 is lower doesn’t imply that it is incorrect, and therefore the implication that steady state would not be rapidly reached would be false. If the small V1 is correct, than the ‘rapid steady state’ using a higher V1 would actually mean there is an (unnecessary) overshoot.

The authors should recognize that Morse and colleagues subsequently published a paper of pharmacokinetic concepts for dexmedetomidine target controlled infusion pumps. Here they demonstrate using their pharmacokinetic model and a model for heart rate and blood pressure that haemodynamic instability can be mitigated with an infusion over 15 minutes, despite a higher dose being required to attain a target effect of sedation compared with a  bolus dose (Morse, Cortinez et al. 2021) .

A: We did add a reference and comment to recognize the simulation study of Morse et al, who confirm the need for longer infusion times to mitigate hemodynamic disturbances. The claim in the Eleveld comment (and in our review)  is not that the Morse model is unsafe under all conditions, the claim is that without limitations to infusion the Morse model could potentially be unsafe, whereas the Hannivoort model would still be within the safe zone for infusion rate if clinically acceptable targets are selected. So by infusing the loading dose over 15 or 30 minutes, or limiting the infusion rate to 6 mcg/kg/h in any case, the hemodynamic side effects can be mitigated indeed. The problem occurs when pump manufacturers do not realize this potential problem, do not limit their pumps, users assume the pump is safe, and a rapid bolus ensues. Two of the authors of this review have seen demonstrations at anaesthesia conferences and business meetings  (not in patients, not commercialized at that time) of pumps with dexmedetomidine models implemented that do not limit the infusion rate, hence our apprehension regarding this topic.

It is also important to realize that with an infusion over a given time, volume is not the sole parameter determining dose. Intercompartment clearances and other compartment volumes are important. 

A: This is true. Therefore, the figures in the Eleveld comment illustrating the doses required to reach a target used the full models for dose estimations, not just V1. We also added a paragraph explaining the problems for V1 quantification in studies in the Morse model paragraph.

I agree with the authors that an infusion rate limit on TCI pumps is important for patient safety.

A: We thank the reviewer for his constructive and detailed comments. It certainly helped to remove some inaccuracies and it improved the flow of the review. We did especially restructure the sections on the Hannivoort en Morse dexmedetomidine models. As such, we are looking forward to your opinion of this reviewed version.